# Metastatic Breast Cancer Recurrence after Bone Fractures

**DOI:** 10.3390/cancers14030601

**Published:** 2022-01-25

**Authors:** Nadia Obi, Stefan Werner, Frank Thelen, Heiko Becher, Klaus Pantel

**Affiliations:** 1Institute for Medical Biometry and Epidemiology, University Medical Center Hamburg-Eppendorf, Martinistr. 52, 20246 Hamburg, Germany; n.obi@uke.de (N.O.); h.becher@uke.de (H.B.); 2Institute for Tumor Biology, University Medical Center Hamburg-Eppendorf, Martinistr. 52, 20246 Hamburg, Germany; st.werner@uke.de; 3Mildred-Scheel-Nachwuchszentrum HaTRiCs4, Universitäres Cancer Center Hamburg, 20246 Hamburg, Germany; 4Analytics & Insights, Techniker Krankenkasse, Bramfelder Str. 140, 22305 Hamburg, Germany; Frank.Thelen@tk.de

**Keywords:** metastasis, breast cancer, risk of relapse, bone fractures, administrative data

## Abstract

**Simple Summary:**

Bone fractures bear potential risk to promote metastatic relapse in breast cancer. We conducted a population-based cohort study of 84,300 breast cancer patients diagnosed between January 2015 and November 2019. Bone fracture after breast cancer diagnosis was associated with an increased metastasis risk. Fractures may pose an increased risk to developing metastasis. Potential clinical implications for cancer patients are in support of fall prevention programs.

**Abstract:**

Experimental studies suggest that bone fractures result in the release of cytokines and cells that might promote metastasis. Obtaining observational data on bone fractures after breast cancer diagnoses related to distant breast cancer recurrence could help to provide first epidemiological evidence for a metastasis-promoting effect of bone fractures. We used data from the largest German statutory health insurance fund (Techniker Krankenkasse, Hamburg, Germany) in a population-based cohort study of breast cancer patients with ICD-10 C50 codes documented between January 2015 and November 2019. The risk of metastasis overall, regional, distant non-bone or bone metastasis related to a fracture was modeled by an adjusted discrete time-to-event analysis with time-dependent exposure. Of 154,000 breast cancer patients, 84,300 fulfilled the inclusion criteria and had a follow-up time of more than half a year. During follow-up, fractures were diagnosed in 13,579 (16.1%) patients. Metastases occurred in 7047 (8.4%) patients; thereof 1544 had affected regional lymph nodes only and 5503 distant metastases. Fractures demonstrated a statistically significant association with subsequent metastasis overall (adjusted HR 1.12, 95% CI 1.04, 1.20). The highest risk for metastasis was observed in patients with subsequent bone metastasis (adjusted HR 1.18, 95% CI 1.05, 1.34), followed by distant non-bone metastasis (adjusted HR 1.16, 95% CI 1.07, 1.26) and lymph node metastasis (adjusted HR 1.08, 95% CI 0.97, 1.21).

## 1. Introduction

Breast cancer is the most commonly diagnosed cancer in women (2.1 million new cases in 2018) and the leading cause of cancer death in women globally (627,000 deaths in 2018) [1]. Metastasis—the spread of tumor cells to distant sites and outgrowth into secondary lesions—is the main cause of cancer-related death in breast cancer and most cancer-related deaths (83% in estrogen receptor (ER)-positive and 87% in ER-negative tumors) happen after distant metastasis formation [2]. Recurrence can occur years after diagnosis and surgical resection of the primary tumor and affects the regional lymph nodes and/or distant organs, such as bone, liver, lungs or brain. In this regard, ER-negative tumors relapse frequently early after diagnosis but the relapse frequency progressively declines over time [2]. In contrast, ER-positive breast cancer recurrences are initially low but continue to occur steadily throughout 20 years after initial diagnosis [2,3]. The German cancer registry Saarland reported 5-year cumulative incidences of distant metastasis in completely resected (R0) tumors of 5.6% in hormone receptor positive breast cancers and 15.9% in HR negative cancers, respectively. In total, the cumulative incidence for distant metastasis was 7.2% [4]. Accumulating evidence suggests that tumor intrinsic genomic alterations are most relevant for determining the risk of tumor recurrence and metastatic relapse [2,5]; on the other hand, external events affecting recurrence and metastatic outgrowth in cancer patients are largely unknown.

The idea that tumor growth and recurrence are evoked by trauma and proximate inflammation or healing processes has existed for more than a century [6,7,8]. This hypothesis is supported by experimental models suggesting a true impact of inflammation [9] and possibly tissue repair [10]. However, these models are hampered by the fact that mice have only a short life span and most tumor models mimic a situation where recurrence occurs within weeks. Thus, although experimental studies provide novel mechanistic insights, they need to be cross-validated by adequate clinical data. To our best knowledge, there is a lack of observational data analyzing whether bone fractures will accelerate or slow down the development of breast cancer recurrence.

In the present study, we therefore tested the hypothesis that bone fractures occurring after initial breast cancer diagnosis might have an impact on the risk of regional and distant breast cancer recurrence. We performed a cohort analysis on 84,300 breast cancer patients using claims data of a statutory health insurance sample in Germany, allowing a follow-up of 5 years.

## 2. Materials and Methods

### 2.1. Study Design, Data Sources and Data Structure

A retrospective register- and population-based cohort study of breast cancer was performed using administrative data for claims purpose of the largest German statutory health insurance fund (Techniker Krankenkasse, TK, Hamburg, Germany). The TK has on average 10 million insured members. The database included records from ambulant and hospital care on selected diagnostic codes (WHO ICD-10 C50 for invasive cancer of the breast but not carcinoma in situ; for all other C codes and M80-M82, see Appendix A) and treatment prescriptions (coded according to international classification for pharmaceutical substances, ATC). These were documented on a quarterly or monthly basis. Sources of diagnoses were flagged as either ambulant or hospital-based, and certainty of ambulant diagnoses had the “status assured” or “status post”. Date variables refer to the year, end of the quarter of ambulant diagnoses, end of the month for the day of discharge from a hospital and month of having filled a prescription, respectively. Clinical data on tumor subtype, stage, death and menopausal status were not available. For this study, the available database included 4,951,968 records from 154,260 women, which had at least one entry with the ICD-10 code C50 (see below for further description of the cohort) from 31 January 2015 to 30 November 2019. All entries of these women were retrieved. The median number of entries per women was 30 (IQR 18–45).

### 2.2. Data Protection and Ethical Considerations

Based on a legal regulation for use of administrative claims data in public health research (§ 75 SGB X), the extracted TK data were anonymized, and included a non-speaking identifier, subject’s birth year and few other variables (see above), so that backtracking of a person is not possible. Therefore, an informed consent was not necessary (see also EU-General data protection regulation, recital 26). This research was conducted according to the principles of the Declaration of Helsinki.

### 2.3. Study Population

From the original cohort of 154,260 individuals we excluded patients as follows (Figure 1): (i) 31,561 patients whose first diagnosis has been tagged with the German ICD-modification “status post” [11], i.e., prevalent cases, (ii) 14,169 patients who were diagnosed for metastasis prior or simultaneously (within one quarter) to the initial BCa diagnosis, (iii) 11,374 patients with a BCa diagnosis in only a single record were excluded due to a potentially false BCa diagnosis, (iv) 657 individuals with endocrine therapy prior to BCa diagnosis, (v) 76 with missing age and (vi) 12,150 patients with a follow-up time of less than six months. Finally, we included 84,300 patients in the main analysis for overall metastasis (Figure 1). The number of patients excluded due to short follow-up varied according to the outcome; it was 10,664 for lymph node metastasis, 7761 for distant bone metastasis and 8689 for non-bone metastasis, respectively.

### 2.4. Statistical Analysis

#### 2.4.1. Exposure Variables

The diagnosis of bone fracture (all ICD-codes for fractures within S and T as well as the code for pathological fractures due to osteoporosis M80) simultaneous or subsequent to BCa diagnosis was considered as time-dependent exposure variable. Those with prior fractures were considered as having had “no fracture”, even if they were diagnosed with a second fracture simultaneous or after BC diagnosis. In total 13,579 of patients were diagnosed with a bone fracture at or after BCa diagnosis.

#### 2.4.2. Outcomes

The primary outcome was a diagnosis of metastases. We further stratified the outcome into three subgroups. (i) Lymph node metastasis, ICD-10 C77, (ii) distant non-bone metastasis, C78 and C79 without C79.5 and (iii) distant bone metastasis, C79.5. The subgroups were non-exclusive. For example, a patient diagnosed both with a lymph node metastasis and with a distant bone metastasis at different or equal times is included in both analyses as a case with corresponding follow-up times. The occurrence of the other metastasis was then ignored.

#### 2.4.3. Other Covariates

The TK dataset included information on potential confounders. These are birth year, source of BCa/fracture/metastasis diagnosis (outpatient care or hospital), year of diagnosis), ICD-codes for diagnosis of other malignant tumors (C00-C97), and a diagnosis of osteoporosis (M81) as well as prescription of anti-estrogens (L02BA), aromatase inhibitors (L02BG), bisphosphonates (M05BA) and bisphosphonate combinations (M05BA). Any depletion of the peripheral estrogen concentration by anti-estrogen treatment, particularly AI, might be associated with a higher risk for osteoporosis [12], which in turn confers a higher risk of fractures. Bisphosphonates, on the other hand, are prescribed as treatments for osteoporosis and preferably in metastatic patients, and may reduce the risk of fractures as well as metastasis [13]. Therefore, bisphosphonates were not included into models. The number of entries per patient per year without entries for fractures in the original TK dataset served as a surrogate for health care use.

#### 2.4.4. Time-to-Event Regression

Time-to-event regression analyses were conducted for each outcome of metastasis with a discrete underlying time scale to account for ties. Hospital date specifications were aligned with ambulant dates by assigning them the end of respective quarter date. Observation time was calculated as date of first entry with BCa diagnosis (earliest date is the first quarter 2015) until date of last entry or first date of metastasis diagnosis, whichever came first. The last occurring date was 30 November 2019. Follow-up time was calculated by the reverse Kaplan–Meyer method. Analyses were adjusted for age at breast cancer diagnosis, source of BCa diagnosis (hospital/ambulant), secondary tumors (time-dependent yes/no), other malignant tumor before BCa diagnosis (yes/no), osteoporosis (time-dependent yes/no), prescriptions of anti-estrogen therapy (time-dependent yes/no) and aromatase inhibitors (time-dependent yes/no). All analyses were stratified by year of diagnosis.

We performed several sensitivity analyses: (i) we excluded all BCa cases that have a first date of diagnoses in 2015, as the majority of these are likely to be prevalent cases. (ii) We minimized the probability of prevalent metastasis in prolonging the waiting period from >0.5 to >1 year between initial breast cancer diagnosis and occurrence of metastasis or censoring. (iii) We repeated (II) for those with a BCa diagnosis after 2015 and (iv) we restricted the analysis to patients with prescriptions of anti-estrogens and AI as an indicator for estrogen receptor positive tumors, which is, in view of prognosis, a more homogeneous group.

Statistical analyses were performed in SAS version 9.4 (SAS Institute, Cary, NC, USA). All tests were two-sided and confidence intervals, not including one, were considered statistically significant.

## 3. Results

### 3.1. Diagnosis of Bone Fractures and Risk of Metastasis

Characteristics of the study population were shown according to the occurrence of a fracture (Table 1). At first BCa diagnosis the median age was 61 years. Patients with a fracture (16.1%) were on average eight years older (median 68 years) than those without a fracture (median 60 years). The majority of fractures were located at lower or upper extremities (49.9%) followed by pathological fractures (18.3%) and spine/pelvis (13.7%), whereas hip fractures occurred rarely in 5.3% (Appendix A). Most patients with fractures had only one specific ICD S-, T- or M80-code (81.2%) (Appendix A). The source of BCa diagnosis was hospital-based in only 9.6% patients with a fracture at or after BCa, versus 17.7% in patients without fractures (Table 1). The proportion of patients with fractures at/after diagnosis decreased over time from 19.3% in 2015 to 1.8% in 2019. In total, the median follow-up time was 4 years and little longer in patients with a fracture (4.25 years). During follow-up, 7047 (8.4%) patients were diagnosed with a metastasis. Patients with any fractures were less likely to be diagnosed with a metastatic relapse compared to those without fractures (7.0% versus 8.6%). However, this difference was lower for a diagnosis of distant bone metastasis. Prevalent or incident second tumors occurred in 15% of all patients and were more common in those with fractures (18.8%). Osteoporosis was diagnosed in 19% without fractures and more than doubled in patients who had a fracture. Anti-estrogens were less frequently prescribed in patients with a fracture compared to those without a fracture, but differences for aromatase inhibitors were small (Table 1).

Unadjusted hazard ratios (HR) for fractures versus no fracture were statistically significantly higher for overall metastasis and distant metastasis, but not for lymph node metastasis (HR 1.03, 95% confidence interval (95% CI) 0.92, 1.14) (Table 2). After adjusting for covariates, the overall risk of a diagnosis of metastasis after the initial BCa was significantly higher in patients who were diagnosed with a bone fracture compared to those without a fracture (adjusted HR 1.12, 95% CI 1.04, 1.20) (Table 2). The highest risk for metastasis was observed in the subgroup of patients with subsequent bone metastasis (HR 1.18, 95% CI 1.05, 1.34), followed by distant non-bone metastasis (HR 1.16, 95% CI 1.07, 1.26) and lymph node metastasis (HR 1.12, 95% CI 1.01, 1.25) (Table 2).

### 3.2. Sensitivity Analysis

After excluding all patients with diagnosis of BCa in 2015, we observed non-significantly elevated HRs for fractures related to distant non-bone (HR 1.28, 95% CI 0.94, 1.75) and bone metastasis (HR 1.45, 95% CI 0.94, 2.24), but not overall or in patients with lymph node metastasis (Table 2). Of note is the change in associations of some covariates with the development of metastasis, e.g., anti-estrogens became protective and source of BCa diagnosis was no more associated to distant metastasis (Table 2, model 3 vs. model 2).

When the waiting period was extended from half a year to more than one year (Appendix A, Model 4), estimates for fractures on distant metastasis were slightly reduced compared to the main Model 2 (Table 2). However, results varied again according to whether patients with BCa diagnosis in 2015 were included or excluded (Appendix A, Model 5). In the latter, HRs for fractures were only non-significantly higher for distant non-bone and, to a lesser extent, bone metastasis.

A similar pattern of associations was observed in the subgroup of patients with endocrine therapy as an indicator for ER-positive tumors (Appendix A, Model 6), i.e., the strongest increase in HRs were observed in distant bone and non-bone metastasis when patients diagnosed in 2015 were included. After exclusion of these patients, fractures were non-significantly associated with non-bone metastasis and significantly associated with distant bone metastasis (HR 1.22, 95% CI 0.80, 1.87; HR 1.78, 95% CI 1.07, 2.95, respectively) (Appendix A, Model 7).

## 4. Discussion

The present study demonstrated that patients diagnosed with a bone fracture after or concurrent with the diagnosis of breast cancer may have an increased chance of being diagnosed with a distant metastasis within 5 years after a BCa diagnosis. Fractures were not consistently related to lymph node metastasis, but they remained associated with higher hazard ratios for metastases in distant organs of the bone and other sites in sensitivity analyses, suggesting a systemic mechanism of action.

Breast cancer cells are frequently present in lymph nodes and distant organs, such as the bone marrow of early-stage breast cancer patients without any clinical or radiological signs of overt metastasis (TNM-stage M0) [14,15,16]. Although these disseminated tumor cells (DTC) pose an increased risk for breast cancer recurrence, approximately 50% of DTC-positive patients do not develop metastasis within 10 years after diagnosis [16,17]. DTCs can survive adjuvant therapy and reside in the bone marrow (and probably other organs) for many years in a stage of “dormancy” [18,19,20,21].

Although the presented epidemiologic data cannot prove any causal relationships, it is tempting to speculate how bone fractures may affect dormant DTCs present in lymph nodes and distant organs. Tissue repair after bone fractures is followed by changes in the immune system (and systemic release of cytokines) [22] that could explain a systemic effect on DTCs even located far away from the fracture site [22,23]. Changes in immune-mediated processes, such as an increase in tumor-promoting M2 macrophages occurring after bone fractures [24,25,26], may promote the growth of DTCs into overt metastases.

Despite the large database and the association found between fractures and metastatic relapse, using a database of a statutory health insurance sample has obvious limitations. In Germany, data need to be deleted after 5 years of storage due to data protection regulations. Our database provides, therefore, no information on late relapses, which are more frequent in hormone receptor-positive patients than HER2-positive or triple-negative patients [3]. It is possible that the present sample included patients with an aggressive disease (e.g., ER-negative patients) over proportional compared to the average population because of the short observation time and the higher need to visit physicians for treatment demands. However, an analysis restricted to patients treated with endocrine therapy as a surrogate for estrogen receptor positivity yielded the same pattern of results as within the total sample. Hence, despite the lack of pathological information, the data do not indicate a major difference if restricted to a positive hormone receptor status. Nevertheless, we cannot exclude that the observed associations are due to chance or unmeasured confounding factors, such as higher tumor stage and heavier treatment, physically non-active lifestyle, medication with anti-inflammatory drugs or Aspirin [27,28] and other inflammatory comorbidities (e.g., fibromyalgia) in those with fractures, all of which were not available in our TK dataset. However, accounting for use of pain-releasing medications in patients with a fracture would have probably diluted the results towards the null.

Health insurance data have been sporadically used for studies of health (care) conditions; some of which performed time-to-event analyses [29,30,31] or annual incidence/mortality calculations [32,33,34]. A general concern with the use of German claims data for epidemiological purposes is that prevalent and incident diagnoses are not distinguishable, leading to an overestimation of the latter, as has been demonstrated for colorectal cancer [10]. According to German cancer registry data we expected between 8750 (2015) and 8375 (2017) incident cases per year in the TK dataset, which was far exceeded in 2015 but not in 2016 and 2017 [35]. We have reduced prevalent breast cancer cases by introducing a one-year look-back period, only including initial C50 diagnoses between 2016 and 2019. Additionally, we extended the minimal follow-up period to reduce the potential for prevalent primary metastasis. These sensitivity analyses demonstrated consistently non-significantly higher estimates for fractures only related to distant metastases. A further limitation is the potential for underreporting of bone metastasis compared to clinical records, as has been demonstrated in a Danish register study [36]. However, we have no evidence that this possible under-reporting was differential in patients with and without fractures. Numbers of entries per patient per year increased with a diagnosis of fracture in the TK dataset, reflecting the patient’s need of additional health care. Thus, the detection of metastasis might have been facilitated after a fracture. On the other hand, the overall 5-year incidence of distant metastases was 7.2% in completely resected (R0) tumors in a German epidemiological cancer registry [4], which compared well to an incidence of 6.5% in our data, in view of a little shorter follow-up and enclosed incomplete resected tumors. Apart from uncertainty in the date of primary diagnosis and date of local relapse, the end of observation time depended on the last date of prescription of treatments or diagnoses of breast cancer, fracture, other malignant tumors, osteoporosis or metastasis. Censoring information due to death or leaving the insurance company were not available in the TK dataset. However, only 12% of all patients had no entry in 2019 whereas in Germany the 5-year absolute mortality was 21% in 2016 [35]. Hence, it is unlikely that the observed association is caused by competing risk of death. However, because of uncertainty of the sequence of events, well-designed prospective studies with longer follow-up and the integration of data on bone fractures in epidemiologic data bases on breast cancer are warranted. Moreover, the effect of fractures in the overall cohort of breast cancer patients was rather moderate and may therefore not counterbalance the advantages of exercise for reducing morbidity and possibly even mortality of cancer patients [37,38], and fall prevention strategies to avoid fractures can be included into these exercise programs.

## 5. Conclusions

In conclusion, the results of our current analysis provide first evidence that fractures may pose an increased risk to develop distant metastasis in breast cancer. Whether this also applies to other bone-seeking tumors, such as prostate cancer, remains to be investigated. A better understanding of the mechanisms behind a potential influence of bone fractures on distant metastatic progression by future experimental studies might lead to the identification of patients with a higher individual risk of metastasis after fractures and the discovery of new preventive strategies focused on these putative high-risk patients to block or slow down metastatic relapse in cancer patients with fractures.

## Figures and Tables

**Figure 1 cancers-14-00601-f001:**
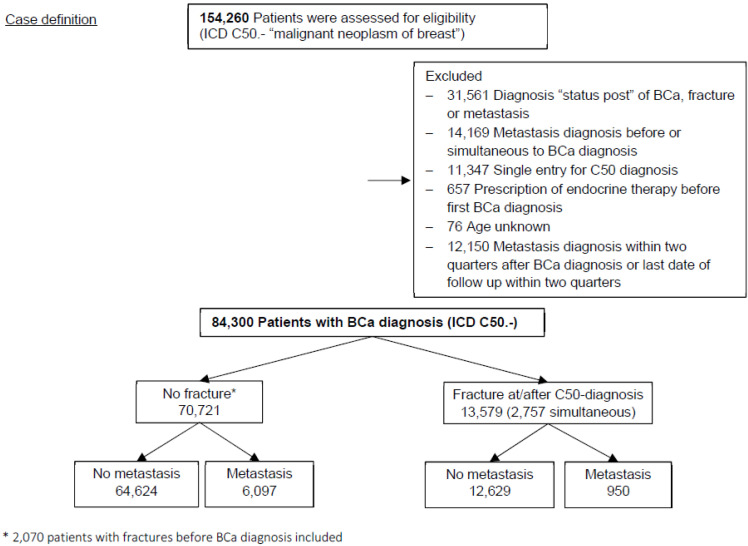
Flowchart of patients included in the analysis of overall metastasis.

**Table 1 cancers-14-00601-t001:** Characteristics of the breast cancer cohort by the occurrence of fractures ^a^.

	Total N = 84,300	No Fracture at/after Diagnosis, N = 70,721	Fracture at/after Diagnosis (ICD-10 S, T & M80), N = 13,579
**Age at diagnosis (years), median (IQR)**	61 (52, 71)	60 (52, 70)	68 (58, 76)
**Source of BCa diagnosis**			
**Hospital care**	13,828 (16.4)	12,527 (17.7)	1301 (9.6)
**Ambulant care**	70,472 (83.6)	58,194 (82.3)	12,278 (90.4)
**Year of BCa diagnosis (col %; row %) ^b^**			
**2015 ^§^**	62,835 (74.5, 100)	50,704 (71.7, 80.7)	12,131 (89.3, 19.3)
**2016**	8197 (9.7, 100)	7406 (10.5, 90.4)	791 (5.8, 9.7)
**2017**	7082 (8.4, 100)	6633 (9.4, 93.7)	449 (3.3, 6.3)
**2018**	6015 (7.1, 100)	5810 (8.2, 96.6)	205 (1.5, 3.4)
**2019**	171 (0.2, 100)	168 (0.2, 98.3)	3 (0.02, 1.8)
**Follow-up-time (years) (median (IQR))**	4.25 (3.00, 4.25)	4.25 (2.75, 4.25)	4.25 (4.00, 4.25)
**Time from diagnosis to fracture (years) (median (IQR))**	-	-	1.50 (0.25, 2.50)
**Metastasis ***			
**No**	77,253 (91.6)	64,624 (91.4)	12,629 (93.0)
**Yes**	7047 (8.4)	6097 (8.6)	950 (7.0)
**Regional**	3226 (3.8)	2865 (4.1)	361 (2.7)
**Distant (not bone)**	4903 (5.8)	4230 (6.0)	673 (5.0)
**Distant bone**	2210 (2.6)	1885 (2.7)	325 (2.4)
**No (2015 cases excluded)**	20,298 (94.6)	18,908 (94.6)	1390 (96.0)
**Yes (2015 cases excluded)**	1167 (5.4)	1109 (5.4)	58 (4.0)
**Regional**	675 (3.1)	650 (3.3)	25 (1.7)
**Distant (not bone)**	632 (2.9)	590 (3.0)	42 (2.9)
**Distant bone**	262 (1.2)	239 (1.2)	23 (1.6)
**Second tumors at/after C50**			
**Yes**	14,606 (15.0)	10,074 (14.2)	2551 (18.8)
**No**	71675 (85.0)	60,647 (85.8)	11,028 (81.2)
**Other tumors prior C50**			
**Yes**	1996 (2.4)	1815 (2.6)	181 (1.3)
**No**	82,304 (97.6)	68,906 (97.4)	13,398 (98.7)
**Osteoporosis**			
**Yes**	19,795 (23.5)	13,429 (19.0)	6366 (46.7)
**No**	64,505 (76.5)	57,292 (81.0)	7213 (53.1)
**Anti-estrogens**			
**Yes**	24,268 (28.8)	21,335 (30.2)	2933 (21.6)
**No**	60,032 (71.2)	49,386 (69.8)	10,646 (78.4)
**Aromatase-inhibitors**			
**Yes**	18,834 (22.3)	15,615 (22.1)	3219 (23.7)
**No**	65,466 (77.7)	55,106 (77.9)	10,360 (76.3)
**Bisphosphonates**			
**Yes**	6678 (7.9)	3986 (5.6)	2692 (19.8)
**No**	77,622 (92.1)	66,735 (94.4)	10,887 (80.2)
**No. of entries per patient year without entries for fractures (median (IQR))**	8.9 (5.4, 12.7)	8.7 (5.4, 12.5)	9.7 (6.1, 13.8)

^a^ N and percentages are shown if not indicated otherwise, ^b^ first available year with C50 entry; * Column % for specified metastases exceed 100%, because an individual may have regional and distant metastasis subsequently or concomitantly; § including prevalent cases. Abbreviations: BCa, breast cancer; IQR, interquartile range.

**Table 2 cancers-14-00601-t002:** Associations between fractures and different subgroups of metastasis in breast cancer (unadjusted and adjusted proportional hazard analysis with time-dependent fractures *).

	Overall Metastasis	LN Metastasis	Distant Non-BM	Distant BM
**N (*n* events)**	84,300 (7047)	84,300 (3226)	84,300 (4903)	84,300 (2210)
	HR (95%-CI)	HR (95%-CI)	HR (95%-CI)	HR (95%-CI)
**Model 1 (unadjusted)**				
**Fractures at/after BCa diagnosis (Reference: none)**	1.14 (1.06, 1.23)	1.03 (0.92, 1.14)	1.20 (1.11, 1.30)	1.21 (1.07, 1.36)
**Model 2 (adjusted)**				
**Fractures at/after BCa diagnosis (Reference: none)**	1.12 (1.04, 1.20)	1.08 (0.97, 1.21)	1.16 (1.07, 1.26)	1.18 (1.05, 1.34)
**Second tumor after BCa**	1.91 (1.81, 2.03)	2.11 (1.94, 2.29)	2.27 (2.12, 2.42)	2.07 (1.88, 2.28)
**Other tumor before BCa**	1.49 (1.27, 1.75)	1.29 (1.01, 1.64)	1.60 (1.30, 1.96)	1.66 (1.22, 2.26)
**Osteoporosis**	1.00 (0.94, 1.07)	0.96 (0.87, 1.05)	0.95 (0.89, 1.02)	1.08 (0.98, 1.20)
**Anti-estrogens**	0.94 (0.89, 0.996)	0.80 (0.73, 0.87)	1.05 (0.99, 1.13)	1.14 (1.04, 1.25)
**Aromatase-inhibitors**	1.27 (1.20, 1.35)	1.28 (1.18, 1.39)	1.43 (1.45, 1.64)	1.87 (1.70, 2.05)
**Source of BCa diagnosis ambulant (Reference: hospital)**	0.90 (0.84, 0.97)	0.96 (0.86, 1.06)	0.90 (0.82, 0.99)	0.98 (0.85, 1.12)
**Age (years)**	0.999 (0.997, 1.001)	0.99 (0.98, 0.99)	1.00 (1.00, 1.01)	0.996 (0.99, 1.00)
**Exclusion of BCa diagnosis in 2015** **N (*n* events)**	21,465 (1167)	21,465 (675)	21,465 (632)	21,465 (262)
**Model 3 (adjusted)**				
**Fractures at/after BCa diagnosis (reference: none)**	0.99 (0.76, 1.30)	0.80 (0.54, 1.19)	1.28 (0.94, 1.75)	1.45 (0.94, 2.24)
**Second tumor after BCa**	1.57 (1.30, 1.89)	1.55 (1.22, 1.98)	1.73 (1.37, 2.19)	2.14 (1.54, 2.98)
**Other tumor before BCa**	1.54 (1.26, 1.88)	1.17 (0.89, 1.57)	1.61 (1.24, 2.09)	1.90 (1.28, 2.80)
**Osteoporosis**	0.93 (0.78, 1.10)	0.998 (0.79, 1.66)	0.85 (0.67, 1.07)	1.12 (0.81, 1.54)
**Anti-estrogens**	0.49 (0.43, 0.57)	0.36 (0.29, 0.44)	0.54 (0.45, 0.66)	0.61 (0.46, 0.81)
**Aromatase inhibitors**	0.74 (0.63, 0.86)	0.62 (0.50, 0.77)	0.79 (0.64, 0.96)	1.13 (0.84, 1.51)
**Source of BCa diagnosis ambulant (Reference: hospital)**	0.93 (0.82, 1.05)	1.01 (0.86, 1.20)	0.85 (0.72, 1.10)	0.95 (0.73, 1.23)
**Age (years)**	0.994 (0.989, 0.999)	0.99 (0.98, 0.99)	1.00 (0.99, 1.01)	0.99 (0.98, 1.01)

* Models were stratified by year of diagnosis (5 or 4 strata) and model 2 and 3 were adjusted for the covariates presented. Most covariates were included as time-dependent variables except of age, source of BCa diagnosis, and other tumors before diagnosis. Reference category is “No”, respectively. Bold: statistically significant exposure estimates. BM, bone metastases; HR, hazard ratio; LN, lymph node.

## Data Availability

Data subject to third party restriction. The data that support the findings of this study are available from Techniker Krankenkasse but restrictions apply to the availability of these data, which were used under license for the current study, and so are not publicly available. Data are, however, available from the authors upon reasonable request and with permission of Techniker Krankenkasse.

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
