# Peer review of "Metastatic Breast Cancer Recurrence after Bone Fractures"

_cancers, 2022, doi:10.3390/cancers14030601_

Round 1

Reviewer 1 Report

This is an interesting study about how fractures may promote breast cancer metastases.

My congratulations to the study team for their work.

Here are my comments:

Material and Methods:

Do you include CDIS in your study? Please specify.

For clinicians, not familiarized with WHO ICD codes a supplemental table with the significance of “all other C codes, M80-M82” may be helpful. Please add this information on supplemental material.

Results

Table 2: Headings are long and trunked, so are a little hard to understand. Please consider abbreviations such as BM for Bone metastases in order to include all the headings in one line.

Is it possible to have information about bone fracture and metastasis localization?

Discussion

Systemic inflammatory diseases such as fibromyalgia are more commons in patients with BC and it may be a confounding bias for the effects of fractures. If identification of these diagnostic is not possible, please include it in the limitations of your study.

Conclusions:

The sentence “However, the effect in the overall cohort of breast cancer patients was rather moderate and may therefore not counterbalance the advantages of exercise for reducing morbidity and possibly even mortality of cancer patients, and fall prevention strategies to avoid fractures can be included into these exercise programs“ is an interesting issue but is not supported by the data of your study.  I suggest developing this interesting topic in the discussion and removing this sentence from the conclusions.

Author Response

Response to Reviewer 1:

This is an interesting study about how fractures may promote breast cancer metastases.

My congratulations to the study team for their work.

Response: Thank you very much for your positive comments.

Here are my comments:

Material and Methods: Do you include CDIS in your study? Please specify.

Response: No, we did not include CDIS (carcinomata ductale in situ), only invasive tumours were included. On page 2, we added this information in Material and Methods in the third sentence:

“The database included records from ambulant and hospital care on selected diagnostic codes (WHO ICD-10 C50 for invasive cancer of the breast but not ductal carcinoma in situ (DCIS), all other C codes, M80-M82, see supplemental Table S1)”.

For clinicians, not familiarized with WHO ICD codes a supplemental table with the significance of “all other C codes, M80-M82” may be helpful. Please add this information on supplemental material.

Response: Yes, we added a supplemental Table S1 and changed the numbers of subsequent supplemental tables (see cited sentence above).

Results:

Table 2: Headings are long and trunked, so are a little hard to understand. Please consider abbreviations such as BM for Bone metastases in order to include all the headings in one line.

Response: We edited the table headings according to the reviewers’ suggestion.

Is it possible to have information about bone fracture and metastasis localization?

Response: In principle, information on localization of metastasis had been available. However, many patients have more than one affected site and numbers tend to be high for unspecific sites (please see the following table). Furthermore, smaller numbers would impair analytic power to detect any association.

Discussion:

Systemic inflammatory diseases such as fibromyalgia are more commons in patients with BC and it may be a confounding bias for the effects of fractures. If identification of these diagnostic is not possible, please include it in the limitations of your study.

Response: We thank the reviewer for this hint and modified the related part of the discussion on page 8 accordingly as follows: “Nevertheless, we cannot exclude that the observed associations are due to chance or unmeasured confounding factors, such as higher tumor stage and heavier treatment, physically non-active lifestyle, medication with anti-inflammatory drugs or Aspirin27, 28, and other inflammatory comorbidities (e.g. fibromyalgia) in those with fractures, all of which were not available in our TK dataset. “ 

Conclusions:

The sentence “However, the effect in the overall cohort of breast cancer patients was rather moderate and may therefore not counterbalance the advantages of exercise for reducing morbidity and possibly even mortality of cancer patients, and fall prevention strategies to avoid fractures can be included into these exercise programs“ is an interesting issue but is not supported by the data of your study.  I suggest developing this interesting topic in the discussion and removing this sentence from the conclusions.

Response: We agree and moved this sentence from the conclusions to the end of the discussion on page 8. “Moreover, the effect of fractures in the overall cohort of breast cancer patients was rather moderate and may therefore not counterbalance the advantages of exercise for reducing morbidity and possibly even mortality of cancer patients 37, 38, and fall prevention strategies to avoid fractures can be included into these exercise programs.

Reviewer 2 Report

The authors in their manuscript detail about a meta-analysis focusing on examining the association of bone fractures with metastases of breast cancer patients.

The cohort of cancer patients involved in the analysis was large. Inclusion criteria for the analyses were appropriate.

The paper is timely, well written and clearly shows the association of bone fractures with higher chance of metastases.

I suggest the authors to add a figure with bar charts (normalized data if possible) to highlight the difference of risks of metastasis between the two main groups (or other groups as well).
